# Social determinants of food group consumption based on Mediterranean diet pyramid: A cross-sectional study of university students

**Roberto Martinez-Lacoba**[1,2,3¤]*, **Isabel Pardo-Garcia**[1,2,3], **Elisa Amo-Saus**[1,3], **Francisco Escribano-Sotos**[1,2,3]

**1** Facultad de Ciencias Económicas y Empresariales, Universidad de Castilla-La Mancha (UCLM), Plaza de la Universidad, Albacete, Spain, **2** Centro de Estudios Sociosanitarios (CESS), Universidad de Castilla-La Mancha (UCLM), Albacete, Spain, **3** Members of the research group Economía, Alimentación y Sociedad (Economics, Food & Society Research Group)

¤ Current address: Facultad de Ciencias Económicas y Empresariales, Universidad de Castilla-La Mancha (UCLM), Plaza de la Universidad, Albacete, Spain
* roberto.mlacoba@uclm.es

**Data Availability Statement:** All relevant data are available from the Dryad repository at https://doi.org/10.5061/dryad.73n5tb2t3.

Considering food habits as a modifiable risk factor, an early intervention on youth people could avoid future health and social costs. We aim to determine the level of compliance with the recommendations of the Mediterranean diet pyramid according to social determinants in university students and to analyse the association of these social determinants (and their interaction with gender) with different food group consumption. We used the records of an electronic cross-sectional survey on university students (n = 593) from inland Spain. The results show, generally, that university students do not fully comply with the recommendations and that gender is the social determinant with the greatest effect on differences in food group consumption. Women have a lower consumption of dairy products, olives, nuts and seeds, red meat, and processed meat, sweets, eggs, alcoholic drinks and fast food; and a higher consumption of fruit, compared with men. Socioeconomic status, geographic area, and whether students cook for themselves have a limited influence on differences in food group consumption, which is inconsistent with the literature. Policy makers should consider this gender gap if they wish to implement a policy based on healthy diet, considering that other social determinants are also important, and could interact with gender.

## Introduction

Economic, cultural, and social resources are known to contribute to the unequal distribution of health outcomes [1], and people with fewer economic resources have shorter life expectancies and suffer more illness than the wealthy [2]. Socioeconomic disparities have been shown to be associated with a greater level of all-cause mortality [3], and although treating current disease is an urgent priority, we should not disregard taking action on the underlying social determinants of health [4]. The literature has evidenced the importance of socioeconomic conditions on health and has demonstrated that socioeconomic adversity is a modifiable risk factor [5,6].

Diet and nutrition are important factors in the promotion and the maintenance of good health throughout the entire life course [7]. A healthy diet helps protect against malnutrition in all its forms as well as a range of noncommunicable diseases, including diabetes, heart

**Funding:** This research has been funded by the University of Castilla-La Mancha Research Group "Economy, Food and Society." (Project 2019-GRIN-27194.). The funder had no role in study design, data collection and analysis, decision to publish, or preparation of the manuscript.

**Competing interests:** The authors have declared that no competing interests exist.

disease, stroke and cancer [8]. However, diet is associated with individual, life-style, social, economic, and geographical factors, among others [9–14]. In other words, social and economic conditions can generate a social gradient in diet quality that contributes to health inequalities [2]. There is evidence showing that adverse childhood and adulthood socioeconomic status in older men is associated with poor diet quality [15]. In addition, most studies have shown that women follow a healthier dietary pattern than men [13,14,16–18], underlining differences in food habits. These inequalities in health–due to gender or material issues–are avoidable [4]; adequate policies could help counterbalance social and cultural behaviour.

In terms of a healthy dietary pattern, Mediterranean diet meets requirements from various perspectives. Mediterranean diet is a healthy dietary pattern that may improve individual health and also obtain social and environmental benefits, among others [19,20], but there is a clear shift away from this food pattern [21]. The westernization of diets -increased intake of meat, fat, processed foods, sugar and salt- is also driven by socioeconomic factors, among other variables [22], and lower-quality diets -usually more economical- tend to be selected by groups of lower socioeconomic status [23].

University students are an important group for the promotion of healthy dietary patterns, because unhealthy lifestyles -including unhealthy diet- are shaped in youth [24–26], and bad habits can compromise health across one's life. There are different determinants of eating behaviour in university students [27]: individual and environmental (physical, social and macro) factors, and even the characteristics of the university. The literature has reported that parental socioeconomic position is associated with children's dietary patterns [14,28], showing that higher parental occupation and education level are associated with higher diet quality [29]. Geographical factors can also interact with others in a complex manner, shaping dietary patterns [8]. Furthermore, young adults usually exhibit bad eating behaviours during the transition from adolescence to adulthood, such as skipping meals (or irregular meal consumption) and frequent snacking, among others, compromising diet quality [30,31]. For this reason, an early intervention in youth through food and health policies could help to combat different social gaps and to reduce future economic burden on health systems.

This work uses a sample of students that was used in an earlier work aiming to study the factors associated with an unhealthy diet [14]. That study analysed diet quality through the use of an index, while the current work has adopted a different approach, using new variables. The aim of this new study is dual. On the one hand, we investigate the level of compliance with the recommendations of the Mediterranean diet pyramid [32] based on individual food group consumption among university students according to social determinants, specifically gender, socioeconomic status, location of the family home, the degree course, and whether the students cook for themselves. On the other hand, we analyse how these social determinants and the interaction with gender may affect the consumption of different food groups, the aim being to illustrate problems related to the intake of these groups, and to encourage the elaboration of specific public policies in this regard.

## Methods

### Design

This study was conducted in the Autonomous Community of Castilla-La Mancha, located in central Spain. Students from the University of Castilla-La Mancha in the cities of Albacete, Ciudad Real, Cuenca, Talavera de la Reina and Toledo participated in the study. We conducted an electronic cross-sectional survey with university students. The design of the study can be consulted in a previous work [14]. The data were collected using the Survey Monkey software [33].

## Participants and environment

A total of 1077 students participated in the study ($n$ = 1077). The final non-probabilistic sample comprised 593 participants (n = 593, 249 men and 344 women). Fig 1 shows the data cleaning process [14]. The information about sample, inclusion and exclusion criteria of participants may be reviewed in our previous work [14].

## Ethics approval and consent to participate

All the students were informed of the aims of the study and participated voluntarily. The completion of the questionnaire was considered to imply informed consent. The study worked with anonymised information. This research was conducted according to the guidelines laid down in the Declaration of Helsinki. The Clinical Research Ethics Committee of the Health Unit of Cuenca certified that the study doesn't need ethics approval according to national guidelines (nr: 2018/P1018).

## Variables included

The survey collected information on demographic (age, gender), socioeconomic (location of family home, parental occupation), and food habit characteristics, among others. Food habit data were collected using a food frequency questionnaire (FFQ) adapted from a questionnaire previously validated in Spanish adult population [34,35]. Participants were asked about their consumption of 141 foods divided into 12 groups: i) dairy products; ii) eggs, meat and fish; iii) vegetables; iv) legumes; v) cereal; vi) oils and fats; vii) fruit; viii) sweets and desserts; ix) beverages; x) spices; xi) precooked products; and xii) fast food (Table A in S1 Supporting Information). Individual foods included in the FFQ can be also seen in Table A in S1 Supporting Information. We readapted these food groups to those in the Mediterranean diet pyramid (Table B in S1 Supporting Information).

**2.3.1. Food group consumption.** The FFQ collected intake frequencies as follows: never or hardly never, one serving per day, 2 to 3 servings per day, 4 to 5 servings per day, 6 or more servings per day, 1 to 2 servings per week, 3 to 4 servings per week, 5 or more servings per week, and 1 to 3 servings per month. We calculated mean daily/weekly servings for each food group. The food groups and recommended consumption are based on the Mediterranean diet pyramid [32]. We added an alcoholic beverage group, with the recommended alcohol intake being based on other studies [36]. Fast food and precooked groups were also considered in the study. We assumed that recommended consumption of these two groups was null. There is evidence that shows fast food consumption has associations with an increased risk of different diseases [37,38]. The composition of the groups can be consulted in Table B in S1 Supporting Information.

**2.3.2. Parental socioeconomic status.** Parental occupations were adapted to the major groups in the International Standard Classification of Occupations (ISCO-08) (one digit) [39], which were then converted to the International Socio-Economic Index of occupational status (ISEI-08) [40]. The ISEI-08 is a continuous variable ranged between 10 and 88. This study considered either the father or mother's occupation, whichever was the higher [41,42]. Self-employed parents were included in ISCO group 5 [43], and unemployed/non-working/retired parents were given the lowest ISEI-08 score (10 points). Mean ISEI scores were calculated when questionnaire occupations fitted two or more ISCO groups. The ISEI-08 was categorised into three groups (low, medium and high socioeconomic status), as follows: ISEI<36 ($n$ = 250); 36≤ISEI<62 ($n$ = 238); and ISEI≥62 ($n$ = 105). This categorised variable is called SES (family's socioeconomic status) in the study. Table C in S1 Supporting Information shows the results of this categorisation.

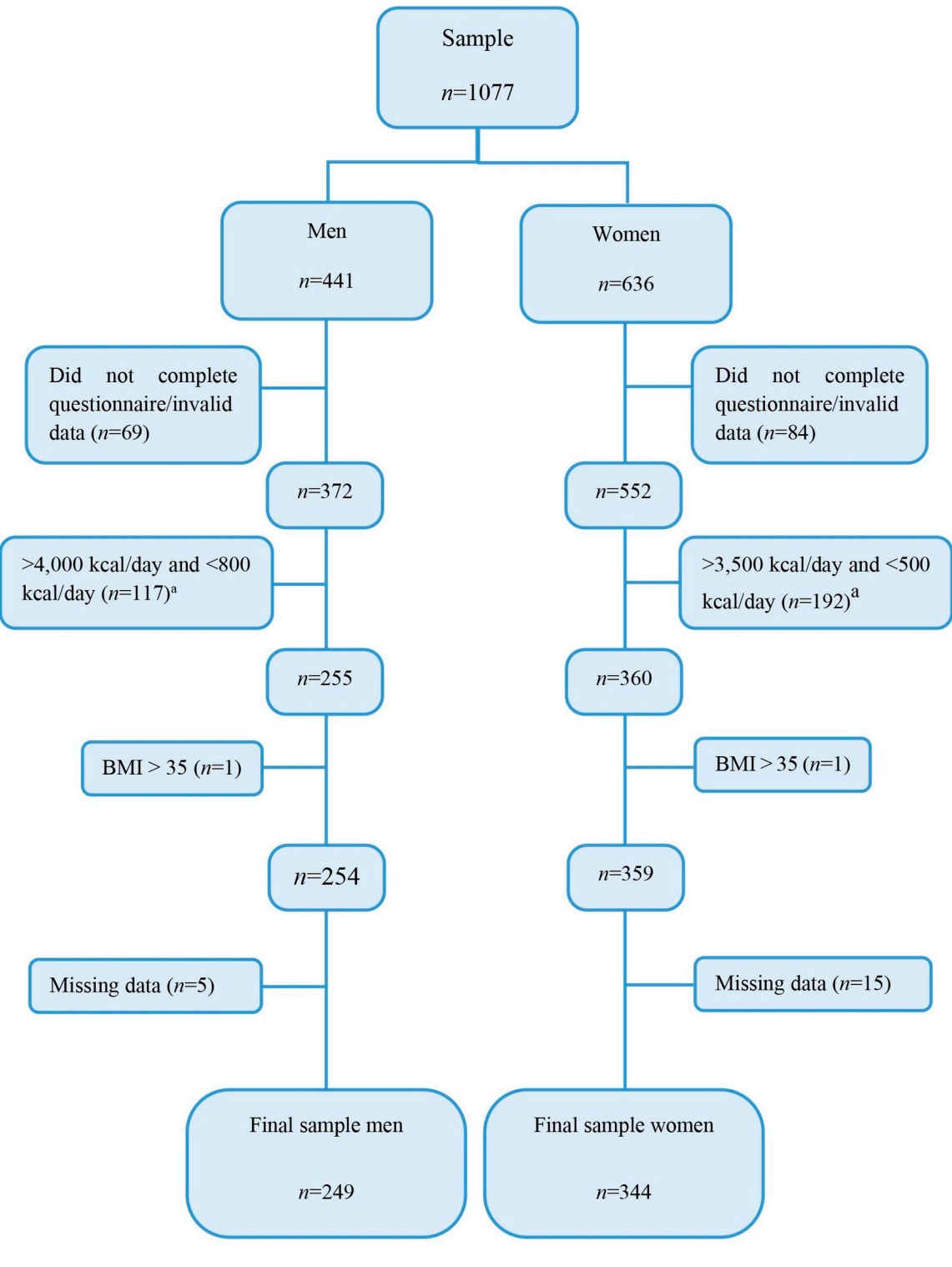

Fig 1. Data cleaning process.

**2.3.3. Family home.** The questionnaire collected family home as follows: village with < 2,000 inhabitants; village with 2,001–5,000 inhabitants; small town with 5,001–15,000 inhabitants; small town with > 15,000 inhabitants; and city. The variable was categorised as follows: rural (village with < 2,000 inhabitants), suburban (village/small town with 2,001–15,000 inhabitants), and urban (small town with > 15,000 inhabitants and city). This categorisation was adapted from other study conducted in Spain [44].

**2.3.4. Student cooks for him or herself or not.** The questionnaire asked whether students cooked for themselves or not. The response was binomial (yes/no). We named this variable CFHS.

**2.3.4. Health or Social Sciences degree.** The questionnaire asked students about the degree course they were enrolled on. We categorised this variable into degrees related to Health Studies or Social Sciences, calling the variable "Degree".

## Missing data analysis

We performed a multiple imputation procedure to deal with missing data, under the missing at random assumption (MAR) [45,46]. We excluded from the missing data analysis participants ($n$ = 153) who: a) did not complete the questionnaire; b) presented invalid data (i.e.: lack of attention, platform failure). We included participants who completed the questionnaire, despite their presenting extreme values (i. e. BMI > 35) or missing values in the data cleaning process. Variables included in the imputation model and results from pooled regression analyses with imputed values are presented in Table D in S1 Supporting Information, Table G in S1 Supporting Information, and Figs B-C in S1 Supporting Information.

## Statistical analysis

For the statistical analysis, we conducted a one-way ANOVA (Welch's ANOVA for unequal variances) and multiple linear regression. The independent variables were gender, family's socioeconomic status, family home, whether the participant cooked for him or herself during the academic year, and the degree course. The dependent variables were food groups. We coded independent variables as dummies in the regression, obtaining the sum of the different comparisons equal to zero [47]. We studied the following comparisons: a) SES: (1) high and medium SES vs. low SES, (2) high SES vs. medium SES; b) family home: (1) urban and suburban vs. rural, (2) urban vs. suburban; c) whether the participant cooked for him or herself during the academic year; d) interaction effects for: gender and SES (1, 2), gender and family home (1, 2), gender and whether the participant cooked for him or herself, and gender and the degree course. Following this regression, we conducted another regression analysis for those dependent variables which presented significant interactions (p<0.10) among the independent ones, excluding independent variables with non-significant effects (p>0.10). The correlations between independent variables were analysed using Spearman's Correlation Coefficient [48,49] and they are shown in Fig A in S1 Supporting Information. The dummy coding is shown in Table E in S1 Supporting Information.

All calculations were made using RStudio [50] and Excel spreadsheet [51].

## Results

Table 1 shows the characteristics of the population by student gender. The students' age, whether they cooked (or not) for themselves, and the degree course were previously shown in our earlier work [14]. In the present study, we also included the location of the family home, but using three categories that consider the size of the family home town. The socioeconomic status of the family is a new variable. The students mean age was 20.21 years (SD = 3.23) and

**Table 1. Characteristics of the study sample.**

|  | All | Males | Females | *P* |
|---|---|---|---|---|
| n (%) | 593 (100) | 249 (41.99) | 344 (58.01) | 0.001 |
| Age (mean, SD) | 20.21 (3.23) | 20.42 (3.21) | 20.06 (3.25) | 0.176 |
| *Family's socioeconomic status (SES) (%)* |  |  |  |  |
| Low | 42.16 | 36.95 | 45.93 | 0.036 |
| Medium | 40.13 | 43.77 | 37.50 | 0.146 |
| High | 17.71 | 19.28 | 16.57 | 0.457 |
| *Family home* |  |  |  |  |
| Rural | 15.51 | 15.26 | 15.70 | 0.976 |
| Suburban | 27.32 | 26.51 | 26.91 | 0.776 |
| Urban | 57.17 | 58.23 | 56.39 | 0.717 |
| *Cooks for him or herself during the academic year (%)* |  |  |  |  |
| Yes | 29.18 | 28.51 | 29.65 | 0.834 |
| No | 70.82 | 71.49 | 70.35 |  |
| *Degree course (%)* |  |  |  |  |
| Health Sciences | 23.90 | 14.10 | 31.10 | <0.001 |
| Social Sciences | 76.10 | 85.90 | 68.90 |  |

Note: Gender related-differences between means or percentages calculated using the *t*-Student test and $\chi^2$.

42% of respondents were male. Regarding socioeconomic status, low and medium SES were the broadest groups in both genders, but the low SES group was larger among women (45.93%). The percentage of students living in an urban area was 57.17%, followed by suburban (27.32%) and rural (15.51%) areas. In addition, the percentage of students cooking for themselves was 29.18%. Finally, the percentage of respondents studying health-related courses was 23.90%, showing significant differences between genders. Fig A in S1 Supporting Information shows the correlations between independent variables. The variables have little correlation ($|\rho| < 0.30$).

Tables 2–6 show mean differences in food group consumption for each social determinant. Table 7 shows and summarises whether students meet recommendations based on the Mediterranean diet pyramid [32]. Finally, Tables 8 and 9 shows results from the multiple regression with complete-case analysis. The results of the multiple linear regression with imputed data can be found in S1 Supporting Information. Figs D-O show the interaction effects between gender and the other social determinants across food groups and Table F in S1 Supporting Information summarises the information on these figures.

## Gender

Table 2 shows mean differences in the number of servings of food groups between men and women. Following the recommendations based on the Mediterranean diet pyramid [32], men failed to comply with the recommendations on olives, nuts and seeds; fruits; vegetables; olive oil; bread, pasta, rice, and other cereals; red meat and processed meat; sweets; and alcoholic beverages (Table 7). Women showed the same habits but complied with recommendations on fruit intake. Men consumed more dairy products, olives, nuts and seeds, red meat and processed food, sweets, eggs, alcohol and fast food compared to women, while women consumed more fruit (Table 2). The multiple regression analysis (Table 8) shows a positive association (p<0.05) between being male and consumption of bread, pasta, rice, other cereals, eggs,

**Table 2. Mean differences in food group consumption by gender (*n* = 593).**

| Food group | | | Gender | | |
|---|---|---|---|---|---|
| | | All | Men (n = 249) | Women (n = 344) | |
| *Daily* | Servings[a] | Mean (SD) | Mean (SD) | Mean (SD) | *P* |
| Dairy products | 2 | 2.93 (2.02) | 3.26 (2.14) | 2.70 (1.90) | <0.001 |
| Olives, nuts, seeds | 1–2 | 0.34 (0.48) | 0.39 (0.50) | 0.31 (0.45) | 0.045 |
| Herbs, spices, garlic, onions | - | 0.57 (0.74) | 0.62 (0.87) | 0.53 (0.63) | 0.153 |
| Fruits | 3–6 | 2.89 (2.34) | 2.66 (1.75) | 3.06 (2.68) | 0.029[†] |
| Vegetables | ≥ 6 | 2.23 (2.40) | 2.08 (2.61) | 2.33 (2.23) | 0.205 |
| Olive oil | 3 | 1.17 (0.90) | 1.10 (0.90) | 1.23 (0.90) | 0.074 |
| Bread, pasta, rice, other cereals | 3–6 | 2.28 (1.44) | 2.36 (1.46) | 2.22 (1.43) | 0.236 |
| *Weekly* | | | | | |
| Potatoes | ≤ 3 | 1.32 (1.49) | 1.35 (1.44) | 1.30 (1.52) | 0.685 |
| Red meat and processed meat | < 2 | 12.90 (9.10) | 14.09 (9.04) | 12.05 (9.05) | 0.007 |
| Sweets | ≤ 2 | 6.39 (6.32) | 7.07 (6.90) | 5.89 (5.83) | 0.026 |
| White meat | 2 | 3.46 (3.35) | 3.68 (3.42) | 3.31 (3.30) | 0.185 |
| Fish, seafood | ≥ 2 | 5.61 (4.35) | 5.19 (4.38) | 5.70 (4.34) | 0.570 |
| Eggs | 2–4 | 2.79 (3.24) | 3.61 (4.28) | 2.20 (2.02) | <0.001[†] |
| Legumes | ≥ 2 | 3.50 (2.75) | 3.75 (2.73) | 3.32 (2.75) | 0.056 |
| *Other food groups of interest* | | | | | |
| Alcoholic drinks (daily) | 1–2 AU/d | 0.70 (1.40) | 0.98 (1.74) | 0.50 (1.05) | <0.001[†] |
| Fast food (weekly) | 0 | 4.48 (2.98) | 4.99 (3.20) | 4.12 (2.76) | <0.001[†] |
| Precooked food (weekly) | 0 | 7.20 (6.11) | 7.12 (4.87) | 7.26 (6.87) | 0.796 |

Abbreviations: AU: Alcohol Units

a. Recommendations based on the Mediterranean diet pyramid and other studies [32,36]

Level of significance of the observed differences between means as assessed by one-way ANOVA or Welch's ANOVA ([†]).

legumes, and fast food. In addition, the fitted model for interaction effects (Table 9) also suggests a positive association between being male and alcohol consumption.

## Parental socioeconomic status

Table 3 shows mean differences in the number of servings of food groups between high, medium and low socioeconomic status (SES). Students comply with the recommendations on dairy products, potatoes, white meat, fish and seafood, eggs and legumes, despite socioeconomic position (Table 3). There is no difference in mean food group consumption across different categories. Students with high or medium SES exhibit a positive association between the consumption of red meat and fast food compared with students with low SES at 0.10 level of significance (Table 8). Interaction effects (Tables 8–9 and F) suggest that men in this social group present higher consumption of servings of bread, pasta rice, other cereals and legumes, and a lower consumption of alcoholic drinks. This means that men with low SES have a higher mean consumption of alcoholic beverages. This is also shown in Table 9 at 0.10 level of significance. Comparing men with high SES and medium SES, men in the former group show a lower consumption of precooked food. Being female with high/medium SES is negatively associated with the consumption of bread, pasta, rice, other cereals and legumes, but female students with high SES (in contrast to medium SES) show a higher consumption of precooked food.

**Table 3. Mean differences in food group consumption by family's socioeconomic status (*n* = 593).**

| Food group | Servings[a] | Socioeconomic status | | | |
|---|---|---|---|---|---|
| | | H | M | L | |
| | | (n = 105) | (n = 238) | (n = 250) | |
| *Daily* | | Mean (SD) | Mean (SD) | Mean (SD) | *P* |
| Dairy products | 2 | 2.77 (1.51) | 2.98 (2.07) | 2.95 (2.16) | 0.663 |
| Olives, nuts, seeds | 1–2 | 0.35 (0.45) | 0.34 (0.47) | 0.34 (0.50) | 0.968 |
| Herbs, spices, garlic, onions | - | 0.50 (0.66) | 0.57 (0.74) | 0.60 (0.76) | 0.531 |
| Fruits | 3–6 | 2.65 (2.07) | 2.92 (2.46) | 2.97 (2.33) | 0.473 |
| Vegetables | ≥ 6 | 2.52 (3.02) | 2.23 (2.53) | 2.10 (1.93) | 0.330 |
| Olive oil | 3 | 1.27 (1.04) | 1.13 (0.86) | 1.18 (0.88) | 0.421 |
| Bread, pasta, rice, other cereals | 3–6 | 2.29 (1.58) | 2.29 (1.45) | 2.27 (1.37) | 0.985 |
| *Weekly* | | | | | |
| Potatoes | ≤ 3 | 1.30 (1.27) | 1.40 (1.43) | 1.26 (1.62) | 0.564 |
| Red meat and processed meat | < 2 | 12.73 (10.01) | 13.93 (9.10) | 12.01 (8.62) | 0.064 |
| Sweets | ≤ 2 | 6.34 (7.08) | 6.37 (6.23) | 6.42 (6.09) | 0.994 |
| White meat | 2 | 3.09 (2.60) | 3.67 (3.54) | 3.42 (3.44) | 0.325 |
| Fish, seafood | ≥ 2 | 5.74 (4.65) | 5.81 (4.21) | 5.37 (4.36) | 0.508 |
| Eggs | 2–4 | 3.04 (4.30) | 2.71 (2.91) | 2.76 (3.03) | 0.677 |
| Legumes | ≥ 2 | 3.35 (2.01) | 3.68 (2.94) | 3.39 (2.82) | 0.422 |
| *Other food groups of interest* | | | | | |
| Alcoholic drinks (daily) | 1–2 AU/d | 0.74 (1.48) | 0.62 (0.99) | 0.78 (1.68) | 0.437 |
| Fast food (weekly) | 0 | 0.67 (0.46) | 0.67 (0.45) | 0.60 (0.39) | 0.163 |
| Precooked food (weekly) | 0 | 1.09 (1.00) | 1.06 (0.74) | 0.98 (0.93) | 0.420 |

Abbreviations: AU: Alcohol Units; H: high; M: medium; L: low; SES: socioeconomic status.

a. Recommendations based on the Mediterranean diet pyramid and other studies (32,36).

Level of significance of the observed differences between means as assessed by one-way ANOVA.

## Location of family home

Table 4 shows mean differences in the number of servings of food groups between students with family homes in urban, suburban or rural areas. On the one hand, there are no differences in compliance with the recommendations depending on the location of the family home, except for fruit consumption in rural area (vs. urban area) (Table 7). There are mean differences in white meat consumption between urban and rural areas. Multiple linear regression results show there are no significant associations between food group consumption and the location of family home at 5% level of significance. At 0.10 level of significance, being from a family living in a rural area is positively associated with the consumption of white meat (compared with urban/rural area), while being from a family living in an urban area shows a positive association with sweet consumption (compared with suburban area).

Interaction effects (Table F) suggest that men from a family living in an urban or suburban area have a lower consumption of fast food; and, at 0.10 level of significance, those from a family living in an urban area (vs. a suburban area) have a higher consumption of bread, pasta, rice, other cereals and red meat and processed meat, and a lower consumption of eggs, but this effect was lost when we fitted the model (Table 9). On the other hand, being female from a family living in an urban or suburban area shows a positive association with the consumption of fast food; and women from a family living in an urban area vs. a suburban area also show a

**Table 4. Mean differences in food group consumption by location of the family home (*n* = 593).**

| Food group | Servings[a] | Family home | | | |
|---|---|---|---|---|---|
| | | U | SU | R | |
| | | (n = 339) | (n = 162) | (n = 92) | |
| *Daily* | | Mean (SD) | Mean (SD) | Mean (SD) | *P* |
| Dairy products | 2 | 3.01 (2.12) | 2.89 (1.88) | 2.72 (1.88) | 0.458 |
| Olives, nuts, seeds | 1–2 | 0.36 (0.49) | 0.28 (0.44) | 0.40 (0.48) | 0.127 |
| Herbs, spices, garlic, onions | - | 0.62 (0.83) | 0.54 (0.62) | 0.45 (0.55) | 0.155 |
| Fruits | 3–6 | 2.92 (2.45) | 2.79 (2.15) | 3.01 (2.26) | 0.746 |
| Vegetables | $\geq 6$ | 2.40 (2.62) | 2.05 (2.33) | 1.90 (1.39) | 0.110 |
| Olive oil | 3 | 1.20 (0.95) | 1.09 (0.82) | 1.25 (0.85) | 0.332 |
| Bread, pasta, rice, other cereals | 3–6 | 2.35 (1.48) | 2.08 (1.31) | 2.37 (1.51) | 0.106 |
| *Weekly* | | | | | |
| Potatoes | $\leq 3$ | 1.35 (1.60) | 1.21 (1.31) | 1.43 (1.36) | 0.455 |
| Red meat and processed meat | < 2 | 12.49 (8.38) | 13.31 (10.74) | 13.73 (8.46) | 0.414 |
| Sweets | $\leq 2$ | 6.68 (6.73) | 5.51 (5.01) | 6.82 (6.73) | 0.116 |
| White meat | 2 | 3.17 (3.15) | 3.65 (3.08) | 4.21 (4.28) | 0.022* |
| Fish, seafood | $\geq 2$ | 5.63 (4.43) | 5.52 (4.49) | 5.68 (3.82) | 0.946 |
| Eggs | 2–4 | 2.64 (2.91) | 2.88 (3.96) | 3.18 (2.99) | 0.343 |
| Legumes | $\geq 2$ | 3.57 (2.98) | 3.21 (2.13) | 3.77 (2.83) | 0.229 |
| *Other food groups of interest* | | | | | |
| Alcoholic drinks (daily) | 1–2 AU/d | 0.69 (1.60) | 0.72 (1.13) | 0.72 (1.02) | 0.970 |
| Fast food (weekly) | 0 | 0.64 (0.42) | 0.62 (0.38) | 0.67 (0.5) | 0.627 |
| Precooked food (weekly) | 0 | 1.03 (0.75) | 1.00 (0.82) | 1.07 (1.28) | 0.828 |

Abbreviations: AU: Alcohol Units; U: Urban; SU: Suburban; R: Rural.

a. Recommendations based on the Mediterranean diet pyramid and other studies (32,36).

Level of significance of the observed differences between means as assessed by one-way ANOVA.

* Significant differences in consumption between urban and rural area (the post-hoc analysis was performed with Tukey Honest Significant Differences).

positive association with the consumption of bread, pasta, rice, other cereals, eggs (effect lost in the fitted model), and a negative association with the consumption of red meat.

## Student cooks for him or herself or not

We analysed whether students cooked for themselves or not yielded differences in the mean number of servings across food groups (Table 5). Students who cook for themselves meet recommendations on fruit consumption, but there are no other differences in meeting recommendations. Participants who cook for themselves have lower consumption of bread, pasta, rice, other cereals and sweets, and higher level of consumption of eggs.

The results of the regression analysis indicate that students who do not cook for themselves show a positive association with the consumption of bread, pasta, rice, and other cereals (at 0.05 level of significance). At 0.10 level of significance, this group shows a positive association with the consumption of sweets. In addition, the results show that students who cook for themselves are positively associated with consumption of eggs. Interaction effects (Table F) show that men who cook for themselves show a positive association with the consumption of potatoes and a higher consumption of white meat and legumes, while women who cook for themselves have a higher consumption of potatoes and a lower consumption of legumes.

**Table 5.  Mean differences in food group consumption depending on whether students cook for themselves or not (*n* = 593).**

| Food group | Servings[a] | CFHS | | |
|---|---|---|---|---|
| | | Yes | No | |
| *Daily* | | Mean (SD) | Mean (SD) | *P* |
| Dairy | 2 | 2.86 (1.91) | 2.97 (2.06) | 0.527 |
| Olives, nuts, seeds | 1–2 | 0.29 (0.36) | 0.37 (0.51) | 0.059 |
| Herbs, spices, garlic, onions | - | 0.56 (0.81) | 0.57 (0.71) | 0.850 |
| Fruits | 3–6 | 3.05 (2.52) | 2.83 (2.26) | 0.312 |
| Vegetables | ≥ 6 | 2.33 (2.96) | 2.18 (2.12) | 0.500 |
| Olive oil | 3 | 1.21 (0.92) | 1.16 (0.9) | 0.601 |
| Bread, pasta, rice, other cereals | 3–6 | 2.07 (1.34) | 2.37 (1.47) | 0.020 |
| *Weekly* | | | | |
| Potatoes | ≤ 3 | 1.43 (1.85) | 1.28 (1.31) | 0.280 |
| Red meat and processed meat | < 2 | 12.55 (9.54) | 13.06 (8.91) | 0.541 |
| Sweets | ≤ 2 | 5.58 (4.81) | 6.72 (6.82) | 0.045 |
| White meat | 2 | 3.79 (3.52) | 3.33 (3.27) | 0.131 |
| Fish, seafood | ≥ 2 | 5.24 (3.92) | 5.76 (4.51) | 0.181 |
| Eggs | 2–4 | 3.32 (4.38) | 2.57 (2.61) | 0.037[†] |
| Legumes | ≥ 2 | 3.25 (2.92) | 3.60 (2.67) | 0.151 |
| *Other food groups of interest* | | | | |
| Alcoholic drinks (daily) | 1–2 AU/d | 0.77 (1.24) | 0.68 (1.46) | 0.484 |
| Fast food (weekly) | 0 | 0.63 (0.44) | 0.64 (0.42) | 0.769 |
| Precooked food (weekly) | 0 | 1.03 (1.11) | 1.03 (0.76) | 0.941 |

Abbreviations: AU: Alcohol Units; CFHS: cooks for him or herself

a. Recommendations based on the Mediterranean diet pyramid and other studies [32,36]

Level of significance of the observed differences between means as assessed by one-way ANOVA or Welch's ANOVA ([†])

## Degree course

Table 6 shows food group consumption by degree course: Health Studies or Social Sciences. Students enrolled on a health-related degree course meet recommendations on fruit consumption, but show the same behaviour as students enrolled in Social Sciences in the rest of food groups. There are significant mean differences in consumption of olive oil, bread, pasta, rice, other cereals, and alcoholic drinks between Health and Social Sciences students, showing that Health Sciences students have a higher consumption of olive oil and cereals, and a lower alcohol consumption. The results of the regression show that students studying health-related courses are positively associated with the consumption of olive oil, bread, pasta, rice and other cereals, and negatively with the consumption of alcohol. Interaction effects suggest that women studying health-related courses have a higher consumption of vegetables, while consumption among their male peers is lower, comparing both analyses with social sciences students.

## Results of multiple imputation

We performed two additional regressions with imputed data (m = 5 and m = 30 subsets). Table G in S1 Supporting Information shows a summary of the results of these regressions compared with complete-case regression at 0.10 level of significance. The comparison partially confirms the results from our complete-case analysis. Regarding gender, the association with the consumption of olives, nuts, and seeds, bread, pasta, rice and others, eggs, and fast food is

**Table 6. Mean differences in food group consumption by degree course (*n* = 593).**

| Food group | Servings[a] | Degree | | P |
| --- | --- | --- | --- | --- |
| | | Health Sciences | Social Sciences | |
| *Daily* | | Mean (SD) | Mean (SD) | *P* |
| Dairy | 2 | 2.85 (1.90) | 2.96 (2.06) | 0.589 |
| Olives, nuts, seeds | 1–2 | 0.36 (0.46) | 0.34 (0.48) | 0.691 |
| Herbs, spices, garlic, onions | - | 0.56 (0.68) | 0.57 (0.76) | 0.878 |
| Fruits | 3–6 | 3.02 (2.05) | 2.85 (2.43) | 0.458 |
| Vegetables | $\geq 6$ | 2.56 (2.11) | 2.12 (2.47) | 0.059 |
| Olive oil | 3 | 1.48 (1.07) | 1.08 (0.82) | <0.001[†] |
| Bread, pasta, rice, other cereals | 3–6 | 2.57 (1.50) | 2.19 (1.41) | 0.005 |
| *Weekly* | | | | |
| Potatoes | $\leq 3$ | 1.39 (1.33) | 1.30 (1.53) | 0.536 |
| Red meat and processed meat | < 2 | 12.07 (8.80) | 13.17 (9.18) | 0.210 |
| Sweets | $\leq 2$ | 5.85 (6.74) | 6.55 (6.18) | 0.251 |
| White meat | 2 | 3.00 (2.07) | 3.61 (3.65) | 0.061 |
| Fish, seafood | $\geq 2$ | 5.59 (3.85) | 5.62 (4.50) | 0.952 |
| Eggs | 2–4 | 2.58 (2.93) | 2.85 (3.33) | 0.388 |
| Legumes | $\geq 2$ | 3.57 (2.82) | 3.48 (2.73) | 0.739 |
| *Other food groups of interest* | | | | |
| Alcoholic drinks (daily) | 1–2 AU/d | 0.29 (0.54) | 0.83 (1.56) | <0.001[†] |
| Fast food (weekly) | 0 | 0.60 (0.40) | 0.65 (0.43) | 0.229 |
| Precooked food (weekly) | 0 | 0.93 (0.75) | 1.06 (0.91) | 0.107 |

Abbreviations: AU: Alcohol Units; CFHS: cooks for him or herself

a. Recommendations based on the Mediterranean diet pyramid and other studies [32,36]

Level of significance of the observed differences between means as assessed by one-way ANOVA or Welch's ANOVA ([†])

confirmed by the three analyses. The association with red meat and legume consumption is confirmed by two analyses. The association with the consumption of fast food is confirmed in the case of families with high or medium socioeconomic status in all analyses, and red meat consumption is confirmed in two of them. The three analyses confirmed white meat consumption (urban/suburban vs. rural areas) and sweet consumption (urban vs. suburban areas). The consumption of bread, pasta, rice, and others, and eggs is confirmed by the three analyses in the cooking habits variable, and sweet consumption in two of them. The three analyses also confirmed the consumption of bread, pasta, rice and other cereals, and alcoholic drinks among health and social science students. As regards interactions, they are wholly confirmed for only two regressions: i) gender and socioeconomic position (high/medium vs. low socioeconomic position) interact with the consumption of bread, pasta, rice, and others; ii) gender and degree course interact with the consumption of olive oil.

## Discussion

This study had two main objectives, which we addressed by means of two analyses. First, we studied the level of compliance with the recommendations of the Mediterranean diet pyramid, stratifying a university sample by five social determinants: gender, socioeconomic status, location of family home, whether the student cooks for him or herself, and the degree course. In this analysis, we included the study of mean differences in food group consumption. Second, we studied differences in food group consumption according to these social determinants and

**Table 7. Compliance with the recommendations of the Mediterranean diet pyramid ($n$ = 593).**

| Food group | | Gender | | SES | | | Family home | | | CFHS | | Degree Course | |
|---|---|---|---|---|---|---|---|---|---|---|---|---|---|
| *Daily* | Servings[a] | M | W | H | Med. | L | U | SU | R | Yes | No | HE | SO |
| Dairy | 2 | Green | Green | Green | Green | Green | Green | Green | Green | Green | Green | Green | Green |
| Olives, nuts, seeds | 1–2 | Red | Red | Red | Red | Red | Red | Red | Red | Red | Red | Red | Red |
| Herbs, spices, garlic, onions | NA | | | | | | | | | | | | |
| Fruits | 3–6 | Red | Green | Red | Red | Red | Red | Green | Green | Green | Red | Green | Red |
| Vegetables | ≥ 6 | Red | Red | Red | Red | Red | Red | Red | Red | Red | Red | Red | Red |
| Olive oil | 3 | Red | Red | Red | Red | Red | Red | Red | Red | Red | Red | Red | Red |
| Bread, pasta, rice, other cereals | 3–6 | Red | Red | Red | Red | Red | Red | Red | Red | Red | Red | Red | Red |
| *Weekly* | | | | | | | | | | | | | |
| Potatoes | ≤ 3 | Green | Green | Green | Green | Green | Green | Green | Green | Green | Green | Green | Green |
| Red meat and processed meat | < 2 | Red | Red | Red | Red | Red | Red | Red | Red | Red | Red | Red | Red |
| Sweets | ≤ 2 | Red | Red | Red | Red | Red | Red | Red | Red | Red | Red | Red | Red |
| White meat | 2 | Green | Green | Green | Green | Green | Green | Green | Green | Green | Green | Green | Green |
| Fish, seafood | ≥ 2 | Green | Green | Green | Green | Green | Green | Green | Green | Green | Green | Green | Green |
| Eggs | 2–4 | Green | Green | Green | Green | Green | Green | Green | Green | Green | Green | Green | Green |
| Legumes | ≥ 2 | Green | Green | Green | Green | Green | Green | Green | Green | Green | Green | Green | Green |
| *Other food groups of interest* | | | | | | | | | | | | | |
| Alcoholic drinks (daily) | 1–2 AU/d | Red | Red | Red | Red | Red | Red | Red | Red | Red | Red | Red | Red |
| Fast food (weekly) | 0 | Red | Red | Red | Red | Red | Red | Red | Red | Red | Red | Red | Red |
| Precooked food (weekly) | 0 | Red | Red | Red | Red | Red | Red | Red | Red | Red | Red | Red | Red |

Abbreviations: AU: Alcohol Units; CFHS: Cooks for him or herself; H: High; HE: Health Studies; L: Low; M: Men; Med.:

Medium, R: Rural, SES: socioeconomic status; SO: Social Sciences SU: Semiurban; U: Urban; W: Women

a. Recommendations based on the Mediterranean diet pyramid and other studies [32,36]

the interaction with gender of socioeconomic status, location of family home, whether the students cook for themselves, and the degree course. To develop this analysis, we performed multiple regression analysis using both complete-case data and imputed data. Our participants had similar ages to those in other studies in university population [31,52–54].

The results from the first analysis indicate, generally, that university students do not fully comply with the recommendations. These results coincide with other studies in the case of fruits and vegetables consumption [55], but not in fish consumption. In addition, in our study female students and participants (both gender) with the family home located in a rural area moderately comply with the recommendations on fruits. Most students, regardless of social determinants, do not comply with the recommendations on daily consumption of olives, nuts and seeds, fruits, vegetables, olive oil, bread, pasta, rice and other cereals, which is consistent with another study [52]. The weekly recommended consumption of red meat and processed meat and sweets is not satisfied, coinciding with the findings of another study using different dietary guidelines [52]. Low compliance with the recommendations of the Mediterranean diet pyramid has been assessed in other studies [56,57], showing that adherence to the Mediterranean dietary pattern is declining among adults and shifting towards a less healthy Western dietary pattern. The loss of the Mediterranean dietary pattern has significant implications in individual health and healthcare systems. It has been widely reported that greater adherence to Mediterranean diet may improve health status [20], and thus promoting the Mediterranean diet is a key point for public health policy not only due to individual health outcomes, but also for its social, economic and environmental benefits [19].

**Table 8. Multiple regression analysis of food groups based on the Mediterranean diet pyramid, social determinants and interactions.**

| Food group | Gender | Socioeconomic position | | Family home | | CFHS | Degree | Interactions | | | | | |
|---|---|---|---|---|---|---|---|---|---|---|---|---|---|
| | | SES (1) | SES (2) | Family home (1) | Family home (2) | | | Gender x SES (1) | Gender x SES (2) | Gender x Family home (1) | Gender x Family home (2) | Gender x CFHS | Gender x Degree |
| *Daily* | | | | | | | | | | | | | |
| Dairy | 0.043 | -0.054 | -0.047 | 0.051 | 0.035 | -0.020 | -0.013 | -0.048 | -0.072 | 0.065 | 0.037 | -0.010 | -0.037 |
| Olives, nuts, seeds | 0.127† | -0.018 | 0.005 | -0.070 | 0.060 | -0.068 | 0.058 | -0.016 | 0.009 | 0.036 | 0.044 | 0.049 | 0.079 |
| Herbs, spices, garlic, onions | 0.064 | -0.060 | -0.033 | 0.067 | 0.059 | 0.013 | 0.002 | 0.043 | 0.017 | -0.025 | 0.056 | -0.003 | -0.003 |
| Fruits | -0.046 | -0.030 | -0.040 | -0.011 | 0.026 | 0.039 | 0.036 | -0.012 | <0.001 | 0.029 | -0.050 | -0.024 | 0.070 |
| Vegetables | -0.100 | 0.057 | 0.045 | 0.060 | 0.067 | 0.066 | 0.025 | 0.042 | 0.013 | 0.005 | 0.031 | 0.024 | -0.104† |
| Olive oil | 0.001 | 0.008 | 0.058 | -0.045 | 0.032 | 0.024 | 0.197*** | -0.043 | 0.041 | -0.006 | -0.017 | -0.03 | 0.062 |
| Bread, pasta, rice, other cereals | 0.140* | 0.007 | <0.001 | -0.053 | 0.033 | -0.087* | 0.130** | 0.118* | 0.041 | 0.010 | -0.077† | 0.015 | 0.017 |
| *Weekly* | | | | | | | | | | | | | |
| Potatoes | -0.009 | 0.028 | -0.026 | -0.031 | 0.045 | 0.039 | 0.042 | -0.009 | -0.032 | 0.001 | 0.018 | -0.115* | 0.064 |
| Red meat and processed meat | 0.120† | 0.083† | -0.050 | -0.039 | -0.045 | -0.020 | 0.004 | 0.053 | 0.016 | 0.052 | 0.081† | 0.033 | 0.072 |
| Sweets | 0.066 | -0.039 | -0.006 | -0.048 | 0.082† | -0.081† | -0.032 | -0.032 | 0.023 | 0.051 | 0.048 | 0.009 | 0.022 |
| White meat | 0.093 | 0.028 | -0.061 | -0.079† | -0.049 | 0.054 | -0.032 | 0.017 | -0.040 | 0.016 | 0.015 | 0.090† | 0.056 |
| Fish, seafood | -0.066 | 0.034 | -0.014 | -0.025 | 0.004 | -0.059 | -0.008 | -0.043 | -0.061 | 0.001 | 0.027 | -0.022 | 0.011 |
| Eggs | 0.272*** | 0.032 | 0.052 | -0.022 | -0.036 | 0.106* | 0.023 | 0.021 | 0.067 | 0.050 | -0.078† | 0.025 | 0.029 |
| Legumes | 0.139* | 0.031 | -0.047 | -0.067 | 0.035 | -0.039 | 0.027 | 0.091* | -0.036 | -0.035 | -0.013 | 0.121* | -0.025 |
| *Other food groups of interest* | | | | | | | | | | | | | |
| Alcoholic drinks | 0.056 | -0.067 | 0.032 | 0.006 | 0.029 | 0.010 | -0.162*** | -0.096* | -0.028 | 0.007 | 0.041 | -0.036 | -0.061 |
| Fast food | 0.182** | 0.079† | 0.003 | -0.059 | 0.008 | -0.010 | -0.021 | 0.049 | -0.005 | -0.099* | -0.016 | -0.022 | 0.007 |
| Precooked food | -0.054 | 0.06 | 0.011 | -0.022 | 0.017 | 0.010 | -0.064 | -0.040 | -0.098* | 0.035 | -0.013 | 0.005 | 0.034 |

Data reported as standardised beta coefficients (β'). Abbreviations: CFHS: Cooks for him or herself; SES: socioeconomic status.

†$P<0.10$;

*$P<0.05$;

**$P<0.01$;

***$P<0.001$

The results of the second analysis indicate that gender is the social determinant with the largest effect on mean differences in food group consumption. Many works have shown that gender is associated with food habits [13,14,16–18,52,54,58], and, as we indicated in the Introduction section, women usually exhibit better food habits than their male counterparts [13,14,16–18]. In the case of male students, our study shows they have a higher intake of dairy products, olives, nuts and seeds, red meat and processed meat, sweets, eggs, alcoholic drinks and fast food. Despite women not complying with most of the Mediterranean diet recommendations, they appear to have healthier dietary patterns than men, according to the literature. The multiple regression analysis confirms these results for the groups of olives, nuts, seeds (at 0.10 level of significance), bread, pasta, rice and others, red meat, eggs, legumes, and fast food, but not for the alcoholic drinks and sweets. This suggests that other variables could influence the consumption of sweets, and alcoholic drinks. In our analysis, we found interactions with socioeconomic position for the alcoholic drink food group. However, the fitted models for

**Table 9. Multiple regression analysis: Fitted models for significant interactions.**

| Food group | Gender | Socioeconomic position | | Family home | | CFHS | Degree | Interactions | | | | | | |
|---|---|---|---|---|---|---|---|---|---|---|---|---|---|---|
| *Daily* | | SES (1) | SES (2) | Family home (1) | Family home (2) | | | Gender x SES (1) | Gender x SES (2) | Gender x Family home (1) | Gender x Family home (2) | Gender x CFHS | Gender x Degree |
| Dairy | - | - | - | - | - | - | - | - | - | - | - | - | - |
| Olives, nuts, seeds | - | - | - | - | - | - | - | - | - | - | - | - | - |
| Herbs, spices, garlic, onions | - | - | - | - | - | - | - | - | - | - | - | - | - |
| Fruits | - | - | - | - | - | - | - | - | - | - | - | - | - |
| Vegetables | -0.102† | - | - | - | - | - | 0.033 | - | - | - | - | - | -0.103† |
| Olive oil | - | - | - | - | - | - | - | - | - | - | - | - | - |
| Bread, pasta, rice, other cereals | 0.113* | 0.001 | - | - | 0.029 | -0.080† | 0.122** | 0.108* | - | - | -0.077† | - | - |
| *Weekly* | | | | | | | | | | | | | |
| Potatoes | -0.036 | - | - | - | - | 0.040 | - | - | - | - | - | -0.128** | - |
| Red meat and processed meat | 0.073† | 0.080† | - | - | -0.043 | - | - | - | - | - | 0.098* | - | - |
| Sweets | - | - | - | - | - | - | - | - | - | - | - | - | - |
| White meat | 0.089* | - | - | -0.085* | - | 0.056 | - | - | - | - | - | 0.080† | - |
| Fish, seafood | - | - | - | - | - | - | - | - | - | - | - | - | - |
| Eggs | 0.239*** | - | - | - | -0.032 | 0.100* | - | - | - | - | -0.074 | - | - |
| Legumes | 0.145* | 0.046 | - | - | - | -0.027 | - | 0.098* | - | - | - | 0.134** | - |
| *Other food groups of interest* | | | | | | | | | | | | | |
| Alcoholic drinks | 0.137** | -0.066 | - | - | - | - | -0.135** | -0.076† | - | - | - | - | - |
| Fast food | 0.177*** | 0.070† | - | -0.055 | - | - | - | - | - | -0.090* | - | - | - |
| Precooked food | -0.036 | - | -0.006 | - | - | - | - | - | -0.087* | - | - | - | - |

Data reported as standardised beta coefficients (β'). Abbreviations: CFHS: Cooks for him or herself; SES: socioeconomic status.

†P<0.10;

*P<0.05;

**P<0.01;

***P<0.001

interactions showed a positive association between being male and alcohol consumption, with the interaction effect being maintained with socioeconomic position. This last analysis also shows a positive association between being female and vegetable consumption and a positive interaction with studying health-related courses. These results in men coincide with other studies in university and adult population for the case of red meat [52,59,60]. Other studies also indicate that men have a higher consumption of alcoholic drinks [53,60,61], eggs [52,61], and sweets [52]. In addition, in the case of female students, fruit consumption is higher than among their male counterparts [52,54,59,61].

Our results indicate that socioeconomic status, geographic area, whether the students cook for themselves, and the degree course have a limited influence on differences in food group consumption. Socioeconomic status shows no differences in any of the food groups, which is inconsistent with the previous literature in adult population [9,62–64]. However, the interaction with gender in the regression analyses show differences in bread, pasta, rice and other

cereals, legumes, alcoholic drinks, and precooked food. Geographical differences, measured by location of the family home -urban, suburban, or rural- have been found for white meat consumption, where students whose families live in rural areas show a higher consumption comparing with urban areas. The limited influence of geographic area has been reported in another study [61]. Moreover, the interaction effect of geographic area with gender shows there are differences in consumption of bread, pasta, rice and others, red meat, eggs and fast food. However, when the fitted models were studied, the interaction of family home with egg consumption was lost. Students who cook for themselves have a lower consumption of bread, pasta, rice and other cereals, sweets, and a higher consumption of eggs. In addition, the interaction of gender with this variable shows differences in consumption of potatoes, white meat and legumes. Studying Health or Social Sciences degree courses shows differences in the consumption of olive oil, bread, pasta, rice, and others, and alcoholic drinks, which are confirmed by the regression analysis. Students of Social Sciences show higher consumption of alcoholic drinks, supported by an article with similar results [65]. The lack of notable differences between the two student profiles (Health and Social Sciences) was unexpected, because a previous work found that studying a non-health related course was associated with an unhealthy diet [14]. This could be explained because that particular work used an index and not the overall consumption of food groups. However, and coinciding with our results, a study on a sample of Health Science students showed that studying health-related courses did not guarantee better choices in food habits [66].

Despite our results being partly inconsistent with other results in the previous literature on social determinants [9,67], they do coincide in the low adherence to Mediterranean diet and we have previously discussed the importance of this dietary pattern. In a previous work [14], 47.90% of students exhibited an unhealthy dietary pattern, which was equivalent to low adherence to Mediterranean diet, and the results of this new work and the earlier one coincide in the importance of developing healthy food habits among university students.

The previous work [14] aimed to analyse the association of individual and social characteristics of the sample with quality of diet, categorising an index of adherence to the Mediterranean diet as healthy/unhealthy diet [36,68]. The previous work [14] aimed to analyse the association between the individual and social characteristics of the sample and quality of diet, categorising an index of adherence to the Mediterranean diet as healthy/unhealthy diet [36,68]. We decided to adopt a different approach in this work because the previous study presented a knowledge gap that we wish to fill. Despite an index usually being a good indicator and showing a general picture of the food habits in the study sample through a global score, it does not clearly show in what food groups decision makers should improve public policies. The previous work studied individual characteristics of the sample (such body mass index), but in this work we focus on social determinants, disregarding the former. For this reason, we followed a different approach in order to show which food groups pose (or not) a problem in the pursuit of a healthier dietary pattern, which could improve long-term health. This new approach may facilitate the elaboration of public policies in some particular groups of students for a specific food group.

Following our results, policy makers should make an effort to promote the Mediterranean diet among university population due to its many benefits [19,20], as students do not fully comply with the recommendations on different food groups. In addition, they should to address the gender gap in the consumption of unhealthy foods, such as sweets, alcoholic drinks and fast food (men showed a higher consumption of these groups), and in healthy foods, such as vegetables or fruits, where men showed a lower consumption, but legumes consumption is higher among men with a better socioeconomic position and who also cook for themselves. Regarding other social determinants, knowledge of healthy food habits should be improved

among Social Science students considering the interaction with gender. Despite our results not showing a substantial association with socioeconomic position, it should be considered since previous literature has shown the influence of socioeconomic status on food habits. In addition, and concerning family home, policy efforts may not be necessary.

This study is not without limitations and the results should be interpreted with caution. First, self-reported food consumption by FFQ can give rise to measurement error [69]. Second, given the characteristics of food frequency questionnaires a memory and social desirability bias might have influenced the results. Third, we could not assess the recommended servings of herbs, spices, garlic and onions because of a lack of information in dietary guideline. A further limitation regards the sample. The final sample represented 4% of the population (593/15,278). Using multiple imputation techniques, we were working with 924 students, who represented 6% of the population. In addition, we did not distinguish between students by year of study (i.e.: first, second- or third-year students).

However, the study has certain strengths. To address FFQ measurement error, we used the criterion of recommended intake in kilocalories, which has no substantial differences from other methods [70,71]. In addition, we dealt with the missing data by using a multiple imputation technique. The multiple regression with the imputed data confirms partially the results from the multiple regression with complete-case analysis and produced comparable standard errors.

The absence of substantive differences by socioeconomic status, geographic area, whether the students cook for themselves, and the degree course could have various explanations. The study population was young and, as noted in a particular study [22], there is a global "nutrition transition" around, which is associated with different diseases and is related to the westernization of diets, and our sample could be affected by these changes. In addition, studies examining socioeconomic disparities usually focus on adult or adolescent populations, and their behaviour may differ from that of a university population. Moreover, university students are a group with special characteristics: small age range, first life stage with more permissive parental control, changes in physical environment, generational socio-cultural norms and values, among others [27]. However, the weak or non-existent in three of four social determinants in our university sample is still important. If policy makers wish to implement a policy based on healthy diet (e.g. Mediterranean diet) in a university population, they must focus their attention on the gender gap (here, the case of women is partially more favourable). Evidently, policy makers should also not forget the social gradient in diet quality. Public policies and health strategies could shape the material conditions of society, helping to improve populations' long-term health.

## Conclusion

This study shows that university students do not fully comply with recommendations on the Mediterranean diet pyramid. In addition, gender is the social determinant with the largest effect on food group consumption. Women have a lower consumption of dairy products, olives, nuts and seeds, red meat, and processed meat, sweets, eggs, alcoholic drinks and fast food, and a higher consumption of fruit, compared with men. Despite our study showing that socioeconomic status, geographic area, and if students cook for themselves have a limited influence on differences in food group consumption, a large body of literature has reported a social gradient in food habits. For this reason, and following our results, in order to avoid future health costs, policy makers should consider the gender gap when implementing policies based on a healthy diet, without forgetting the importance of the other social determinants.

## Supporting information

**S1 Supporting information. Supporting information.** This file includes:

- Multiple regression outputs of complete-case analysis, and multiple imputation analyses (m = 5 and m = 30 subsets).

- Table A. Foods and food groups collected in questionnaire.

- Table B. Food groups in the Mediterranean diet pyramid and foods from the FFQ

- Table C. Occupations collected in the questionnaire: ISCO, ISEI-08 and SES

- Table D. Variables sorted by percentage of missing

- Table E. Independent variables: dummy coding

- Table F. Summary of interactions between gender and the other social determinants across food groups

- Table G. Results from complete-case and imputed data regressions

- Fig A. Correlation across independent variables

- Fig B. Density plots of food groups after imputation of values: complete-case analysis and multiple imputation (m = 5)

- Fig C. Density plots of food groups after imputation of values: complete-case analysis and multiple imputation (m = 30)

- Fig D. Interaction effect between SES (1) and gender in the food group "Bread, pasta, rice, and other cereals" (*n* = 593)

- Fig E. Interaction effect between SES (1) and gender in the food group "Legumes" (*n* = 593)

- Fig F. Interaction effect between SES (1) and gender in the food group "Alcoholic drinks" (*n* = 593)

- Fig G. Interaction effect between SES (2) and gender in the food group "Precooked" (*n* = 593)

- Fig H. Interaction effect between family home (1) and gender in the food group "Fast food" (*n* = 593)

- Fig I. Interaction effect between family home (2) and gender in the food group "Bread, pasta, rice, and other cereals" (*n* = 593)

- Fig J. Interaction effect between family home (2) and gender in the food group "Red meat and processed meat" (*n* = 593)

- Fig K. Interaction effect between family home (2) and gender in the food group "Eggs" (*n* = 593)

- Fig L. Interaction effect between CFHS and gender in the food group "Potatoes" (*n* = 593)

- Fig M. Interaction effect between CFHS and gender in the food group "White meat" (*n* = 593)

- Fig N. Interaction effect between CFHS and gender on Legumes food group (*n* = 593) (DOCX)

## Acknowledgments

The authors thank all professors and colleagues for their time spent. We specially thank Miguel Ángel Gómez Borja for helping us with the survey software.

## Author Contributions

**Conceptualization:** Roberto Martinez-Lacoba, Isabel Pardo-Garcia, Elisa Amo-Saus, Francisco Escribano-Sotos.

**Data curation:** Roberto Martinez-Lacoba, Isabel Pardo-Garcia, Elisa Amo-Saus, Francisco Escribano-Sotos.

**Formal analysis:** Roberto Martinez-Lacoba, Isabel Pardo-Garcia, Elisa Amo-Saus, Francisco Escribano-Sotos.

**Investigation:** Roberto Martinez-Lacoba, Isabel Pardo-Garcia, Elisa Amo-Saus, Francisco Escribano-Sotos.

**Methodology:** Roberto Martinez-Lacoba, Isabel Pardo-Garcia, Elisa Amo-Saus, Francisco Escribano-Sotos.

**Resources:** Roberto Martinez-Lacoba, Isabel Pardo-Garcia, Elisa Amo-Saus, Francisco Escribano-Sotos.

**Software:** Roberto Martinez-Lacoba.

**Supervision:** Roberto Martinez-Lacoba, Isabel Pardo-Garcia, Elisa Amo-Saus, Francisco Escribano-Sotos.

**Validation:** Roberto Martinez-Lacoba, Isabel Pardo-Garcia, Elisa Amo-Saus, Francisco Escribano-Sotos.

**Visualization:** Roberto Martinez-Lacoba, Isabel Pardo-Garcia, Elisa Amo-Saus, Francisco Escribano-Sotos.

**Writing – original draft:** Roberto Martinez-Lacoba, Isabel Pardo-Garcia, Elisa Amo-Saus, Francisco Escribano-Sotos.

**Writing – review & editing:** Roberto Martinez-Lacoba, Isabel Pardo-Garcia, Elisa Amo-Saus, Francisco Escribano-Sotos.

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
