## [Decision Letter · Decision Letter 0]

2 Aug 2019

PONE-D-19-14439

Social determinants of food group consumption based on Mediterranean diet pyramid: a cross-sectional study of university students

PLOS ONE

Dear Dr. Roberto Martínez Lacoba,

Thank you for submitting your manuscript to PLOS ONE. After careful consideration, we feel that it has merit but does not fully meet PLOS ONE’s publication criteria as it currently stands. Therefore, we invite you to submit a revised version of the manuscript that addresses the points raised during the review process.

In your revised version, you need address all the recomendations made by the Reviewer 1 and also added more background about differences in eating habits according to gender in the introducción. In addition,  please note that the present submission is closely related to a previously published work by you: "Socioeconomic, demographic and lifestyle-related factors associated with unhealthy diet: a cross-sectional study of university students". In this regadr, PLOS ONE criteria on related manuscripts (http://journals.plos.org/plosone/s/submission-guidelines#loc-related-manuscripts) require that related studies are adequately mentioned in the present submission (as you did), but PLOS ONE also require that the rationale of these separate analyses is clearly discussed, and the differences between the two works are clearly illustrated. So, you must address this issue in a proper way in your revised version.

We would appreciate receiving your revised manuscript by October 31 2019. To enhance the reproducibility of your results, we recommend that if applicable you deposit your laboratory protocols in protocols.io, where a protocol can be assigned its own identifier (DOI) such that it can be cited independently in the future. For instructions see: http://journals.plos.org/plosone/s/submission-guidelines#loc-laboratory-protocols

We look forward to receiving your revised manuscript.

Kind regards,

Berta Schnettler

Academic Editor

PLOS ONE

Journal Requirements:

3. Please ensure that the findings of your previous analysis "Socioeconomic, demographic and lifestyle-related factors associated with unhealthy diet: a cross-sectional study of university students" are thoroughly discussed in the introduction and Discussion, to allow a broad overview of the field, and to illustrate the background of the present analysis.

6. We noted in your submission details that a portion of your manuscript may have been presented or published elsewhere. [Yes. Figure 1 is the same we used in our paper entitled: "Socioeconomic, demographic and lifestyle-related factors associated with unhealthy diet: a cross-sectional study of university students". The inclusion of this figure does not constitute dual publication because it only shows the data cleaning process and it does not compromise the results.]

Please clarify whether this  publication was peer-reviewed and formally published. If this work was previously peer-reviewed and published, in the cover letter please provide the reason that this work does not constitute dual publication and should be included in the current manuscript.

Reviewers' comments:

Reviewer's Responses to Questions

**Comments to the Author**

1. Is the manuscript technically sound, and do the data support the conclusions?

Reviewer #1: Partly

Reviewer #2: Yes

2. Has the statistical analysis been performed appropriately and rigorously? 

Reviewer #1: N/A

Reviewer #2: Yes

3. Have the authors made all data underlying the findings in their manuscript fully available?

Reviewer #1: Yes

Reviewer #2: Yes

4. Is the manuscript presented in an intelligible fashion and written in standard English?

Reviewer #1: Yes

Reviewer #2: Yes

5. Review Comments to the Author

Reviewer #1: Dear corresponding author,

It was my pleasure to review your study. The paper broadly seems well written and it is welcome the analysis of social determinants associated with the quality of the diet. However, I do have some

recommendations, which are described below:

Introduction:

You should include more background on risk behaviors in students that trigger an unhealthy diet, e.g. skipping meals frequently, eating between meals and having a high consumption of ultra-processed food.

Methods:

Participants: It is important to mention that it is a non-probabilistic sample, for convenience instead of a representative sample.

Variables included: According to several studies, the importance of the degree course variable (Health Sciences, Social Sciences) has been communicated, together with its conclusion in its article: “Socioeconomic, demographic and lifestyle-related factors associated with unhealthy diet”, BMC Public Health 2018, that said: “…. finally, not studying a health-related course are the factors associated with a lower quality diet ”. This variable must be included in your analyzes.

Why is it mentioned that 141 foods were divided into 12 groups in the FFQ? And then, in the analysis of the data (tables) 17 food groups appear. This must be clarified.

Stadistical analysis: The analysis shown in Table 3 and 4 about mean differences in food group consumption by family´s socioeconomic status or the location of the family home, respectively. They cannot present two independent analyzes for the same variable; this increases the type 1 error. Therefore, I would suggest you to work the three respective levels for each variable, despite the fact that in the multiple regression you worked with the dummies variables. You should perform also ANOVA with multiple comparisons in case you show significant differences.

The table 7 shows multiple regression analysis of food groups, social determinants and interaction. More precision is lacking in the final model, considering the interactions that were significant. I suggest you to generate different models for the combinations of significant interactions, with this you can really respond to the second objective of your paper that says: “we analyze how these social determinants and the interaction with gender may affect different food group consumption.”

Discussion

In relation with the limitations of the study, you should mention that the final sample only represented 4% (593 / 15,278) of the population. In addition, there is no clarity of the level that students take, because it is not the same reality to be a freshman in comparison with those of higher courses.

Reviewer #2: This paper is very correct in terms of methodology, has an adequate sample size, but the principal debilitie that present is about the theme investigated, because is not novedous. However, i think that paper can be a good input to actualizated the information avaliable in this field of research, specially to define interventions and/or public policies to prevent the worsening of eating habits in emerging adults of the Spanish state

6. PLOS authors have the option to publish the peer review history of their article (what does this mean?). If published, this will include your full peer review and any attached files.

Reviewer #1: No

Reviewer #2: No

---

## [Author Response · Author response to Decision Letter 0]

7 Nov 2019

Dear Editor,

Thank you for your comments and the opportunity to improve our paper, and we thank the reviewers for their work. We consider that the proposed changes improve the scientific quality of our manuscript. The changes have been highlighted. We have organized our response as follows. First, we have responded to the suggestions of the academic editor and the changes in the main manuscript have been highlighted in blue. Secondly, we have made the changes suggested by the journal (highlighted in blue in the main manuscript), and then we have made the corrections suggested by the reviewers (highlighted in yellow in the main manuscript). 

Responses to academic editor

1. In your revised version, you need (…) add more background about differences in eating habits according to gender in the introduction.

Thank you for your suggestion. We have added more background about differences in eating habits according to gender in the introduction. This can be seen on page 2 (lines 39-42).

2. In addition, please note that the present submission is closely related to a previously published work by you: "Socioeconomic, demographic and lifestyle-related factors associated with unhealthy diet: a cross-sectional study of university students". In this regard, PLOS ONE criteria on related manuscripts require that related studies are adequately mentioned in the present submission (as you did), but PLOS ONE also require that the rationale of these separate analyses is clearly discussed, and the differences between the two works are clearly illustrated.

Thank you for the observation. Please, if we misunderstood your comment, let us know. 

The differences between our previous work entitled “Socioeconomic, demographic and lifestyle-related factors associated with unhealthy diet: a cross-sectional study of university students” and this new work are as follows. First, our previous article’s aim was to understand what factors were associated with unhealthy diet (defining this categorical variable by using an index of adherence to Mediterranean diet (1,2)) (as discrete variable) while the new work has used the recommendations of food group consumption based on the Mediterranean diet pyramid (3), grouping foods using this pyramid. This means that in our previous work we divided the sample, in some way, into two groups (unhealthy diet vs. healthy diet) and in the new paper we studied the level of compliance with the recommendations of Mediterranean diet. We wanted to show what food groups are a problem (or not) in our sample by separately studying food groups. An index is usually a good indicator and shows a general picture of the problem (in other words, you could know whether an individual gets a good or bad score), but it does not show in what food group you should improve public policies. In addition, we focused on social determinants, disregarding the individual characteristics of the university students. With our new work, we now know, for example, that students must improve vegetable consumption, red meat and processed meat consumption or alcoholic drink consumption. 

Following your recommendations, we have changed the last paragraph of the Introduction, as can be seen on pages 3-4 (lines 64-75). In addition, we have also included some reflections related to these suggestions in the Discussion section (page 23, lines 343-346 and 348-350; pages 24-25, lines 381-391; page 25, lines 395-398; pages 25-26, lines 399-409).

1. Sofi F, Macchi C, Abbate R, Gensini GF, Casini A. Mediterranean diet and health status: an updated meta-analysis and a proposal for a literature-based adherence score. Public Health Nutr [Internet]. 2013;17(12):2769–82. Available from: https://doi.org/10.1017/S1368980013003169

2. Sofi F, Dinu M, Pagliai G, Marcucci R. Validation of a literature-based adherence score to Mediterranean diet: the MEDI-LITE score. Int J Food Sci Nutr [Internet]. 2017;68(6):757–62. Available from: http://dx.doi.org/10.1080/09637486.2017.1287884

3. Bach-Faig A, Berry EM, Lairon D, Reguant J, Trichopoulou A, Dernini S, et al. Mediterranean diet pyramid today. Science and cultural updates. Public Health Nutr [Internet]. 2011;14(12A):2274–84. Available from: https://doi.org/10.1017/S1368980011002515

3. Thank you. We have changed our funding section on page 29 (lines 480-481).

Review comments to the Author (Reviewer 1)

 Thank you very much for your comments and suggestions. We think your review improves the quality of our manuscript and also clarifies some important points. All your suggestions are highlighted in yellow.

1. Introduction: You should include more background on risk behaviors in students that trigger an unhealthy diet, e.g. skipping meals frequently, eating between meals and having a high consumption of ultra-processed food.

Following your suggestion, we have included more background about risk behaviours in students in page 3 (lines 58-61).

2. Methods

i. Participants: It is important to mention that it is a non-probabilistic sample, for convenience instead of a representative sample.

Thank you. Following your advice, we have added the required information on page 4 (lines 84-85). 

ii. According to several studies, the importance of the degree course variable (Health Sciences, Social Sciences) has been communicated, together with its conclusion in its article: “Socioeconomic, demographic and lifestyle-related factors associated with unhealthy diet”, BMC Public Health 2018, that said: “…. finally, not studying a health-related course are the factors associated with a lower quality diet ”. This variable must be included in your analyzes.

Thank you for your suggestion. Following your recommendation, we have added the variable “degree course”. A paragraph was added in Methods section (page 6, lines 135-138), Results section (pages 20-21, lines 278-291) and Discussion section (pages 24-25, lines 380-392). We have changed “Table S5. Independent variables: dummy coding”, and Table 1 adding the new information. We have created Table 6. The results of the regression analysis have also changed.

Please, if you have any further suggestions, let us know.

iii. Why is it mentioned that 141 foods were divided into 12 groups in the FFQ? And then, in the analysis of the data (tables) 17 food groups appear. This must be clarified.

Thank you. This was unclear. We have revised this point and changed the text on page 5 (lines 98-99) and we have modified Table S1.

iv. The analysis shown in Table 3 and 4 about mean differences in food group consumption by family´s socioeconomic status or the location of the family home, respectively. They cannot present two independent analyzes for the same variable; this increases the type 1 error. Therefore, I would suggest you to work the three respective levels for each variable, despite the fact that in the multiple regression you worked with the dummies variables. You should perform also ANOVA with multiple comparisons in case you show significant differences.

Thank you. Following your suggestion, we have performed the ANOVA analysis with the three respective levels for each variable. The changes have been introduced in Table 3 and Table 4, and the related results rewritten. 

v. The table 7 shows multiple regression analysis of food groups, social determinants and interaction. More precision is lacking in the final model, considering the interactions that were significant. I suggest you to generate different models for the combinations of significant interactions, with this you can really respond to the second objective of your paper that says: “we analyze how these social determinants and the interaction with gender may affect different food group consumption.”

Thank you for your comment. Following your suggestion, we have added a new table (Table 9) including this. This table is entitled “Multiple regression analysis: fitted models for significant interactions”. The analysis was performed including all the significant interactions and significant independent variables. 

However, if we have misunderstood your suggestion, please, let us know.

3. Discussion

i. In relation with the limitations of the study, you should mention that the final sample only represented 4% (593 / 15,278) of the population. 

Thank you, we have added the suggested information. We also added this because we were using multiple imputation techniques, working with 924 students. You can find that on page 26 (lines 415-417). 

ii. In addition, there is no clarity of the level that students take, because it is not the same reality to be a freshman in comparison with those of higher courses.

Thank you. Following your recommendation, we have written this information as a limitation of the study (page 26, lines 417-419). 

 

Review comments to the Author (Reviewer 2)

Thank you very much for your comments and review.

---

## [Decision Letter · Decision Letter 1]

12 Dec 2019

PONE-D-19-14439R1

Social determinants of food group consumption based on Mediterranean diet pyramid: a cross-sectional study of university students

PLOS ONE

Dear Dr. Martinez-Lacoba

Thank you for submitting your manuscript to PLOS ONE. After careful consideration, we feel that it has merit but does not fully meet PLOS ONE’s publication criteria as it currently stands. Therefore, we invite you to submit a revised version of the manuscript that addresses the points raised during the review process.

Your manuscrip submited to Plos One have  valuable information, which was not at all covered in the previous submission; and the Result section seems to contain very interesting data. However, in my opinion and on the basis of the opinion of one of the Reviewers, the difference between this manuscript and your previously published manuscript is not well illustrated in the Introduction and in the Discussion (in particular, the first paragraph of the discussion in which it seems to mix together the two studies).

Therefore, you need to provide a better rationale for this study, and carefully explain in the Discussion (and not only in your response to reviewers) what insights were found by your new analysis, without conflating results that were already presented in the previous manuscript. Moreover, the introduction needs to make clear that the same sample was analysed in a previous article, and that the data of Table 1 (where the characteristics of the study sample are reported) have been previously shown.

We would appreciate receiving your revised manuscript by January the 2nd, 2020. To enhance the reproducibility of your results, we recommend that if applicable you deposit your laboratory protocols in protocols.io, where a protocol can be assigned its own identifier (DOI) such that it can be cited independently in the future. For instructions see: http://journals.plos.org/plosone/s/submission-guidelines#loc-laboratory-protocols

We look forward to receiving your revised manuscript.

Kind regards,

Berta Schnettler

Academic Editor

PLOS ONE

Reviewers' comments:

Reviewer's Responses to Questions

**Comments to the Author**

1. If the authors have adequately addressed your comments raised in a previous round of review and you feel that this manuscript is now acceptable for publication, you may indicate that here to bypass the “Comments to the Author” section, enter your conflict of interest statement in the “Confidential to Editor” section, and submit your "Accept" recommendation.

Reviewer #1: All comments have been addressed

Reviewer #2: All comments have been addressed

2. Is the manuscript technically sound, and do the data support the conclusions?

Reviewer #1: Yes

Reviewer #2: Yes

3. Has the statistical analysis been performed appropriately and rigorously? 

Reviewer #1: Yes

Reviewer #2: Yes

4. Have the authors made all data underlying the findings in their manuscript fully available?

Reviewer #1: Yes

Reviewer #2: Yes

5. Is the manuscript presented in an intelligible fashion and written in standard English?

Reviewer #1: Yes

Reviewer #2: Yes

6. Review Comments to the Author

Reviewer #1: Dear correspondent author, I agree with the new version of the article. All comments have been addressed. Thank you.

Reviewer #2: Even when this paper is very correct with the norms of the journal, i decide to reject that submission because is very similar with another works already published in this field, even of the same authors, and, for my perspective, dont give nothing really new of this research ambit.

7. PLOS authors have the option to publish the peer review history of their article (what does this mean?). If published, this will include your full peer review and any attached files.

Reviewer #1: No

Reviewer #2: No

---

## [Author Response · Author response to Decision Letter 1]

18 Dec 2019

Rebuttal letter including responses to academic editor and reviewers

Thank you for your comments and the opportunity to improve our paper, and we thank the reviewers for their work, once again. The changes have been highlighted. 

On this occasion, we have responded to your suggestion divided into three paragraphs and the changes in the main manuscript have been highlighted in yellow. 

Responses to academic editor

1. “Your manuscrip submited to Plos One have valuable information, which was not at all covered in the previous submission; and the Result section seems to contain very interesting data.”

Thank you very much.

2. “However, in my opinion and on the basis of the opinion of one of the Reviewers, the difference between this manuscript and your previously published manuscript is not well illustrated in the Introduction and in the Discussion (in particular, the first paragraph of the discussion in which it seems to mix together the two studies). Therefore, you need to provide a better rationale for this study, and carefully explain in the Discussion (and not only in your response to reviewers) what insights were found by your new analysis, without conflating results that were already presented in the previous manuscript.” 

 Thank you for your suggestion. 

We have simplified the first paragraph of the Discussion section in order to improve the precision and it now read (page 23, line 329-330): “This study had two main objectives, which we addressed by means of two analyses. First, we studied the level of compliance (…)”. On the other hand, in the Discussion section (page 26, lines 416-424), we have added an explanation which illustrates the difference between our previous work and this new one. It now reads as follows (we have underlined the explanation): “The previous work [14] aimed to analyse the association between the individual and social characteristics of the sample and quality of diet, categorising an index of adherence to the Mediterranean diet as healthy/unhealthy diet [36,68]. We decided to adopt a different approach in this work because the previous study presented a knowledge gap that we wish to fill. Despite an index usually being a good indicator and showing a general picture of the food habits in the study sample through a global score, it does not clearly show in what food groups decision makers should improve public policies. The previous work studied individual characteristics of the sample (such body mass index), but in this work we focus on social determinants, disregarding the former. For this reason, we followed a different approach in order to show which food groups pose (or not) a problem in the pursuit of a healthier dietary pattern, which could improve long-term health”. 

 In addition, we added a new paragraph in the Discussion section (page 27, lines 429-441) trying to show the insights of our work and suggesting recommendations to decision/policy makers: “Following our results, policy makers should make an effort to promote the Mediterranean diet among university population due to its many benefits [19,20], as students do not fully comply with the recommendations on different food groups. In addition, they should to address the gender gap in the consumption of unhealthy foods, such as sweets, alcoholic drinks and fast food (men showed a higher consumption of these groups), and in healthy foods, such as vegetables or fruits, where men showed a lower consumption, but legumes consumption is higher among men with a better socioeconomic position and who also cook for themselves. Regarding other social determinants, knowledge of healthy food habits should be improved among Social Science students considering the interaction with gender. Despite our results not showing a substantial association with socioeconomic position, it should be considered since previous literature has shown the influence of socioeconomic status on food habits. In addition, and concerning family home, policy efforts may not be necessary.”

Please, if you have any other suggestion or you think we should improve precision, let us know.

3. “Moreover, the introduction needs to make clear that the same sample was analysed in a previous article, and that the data of Table 1 (where the characteristics of the study sample are reported) have been previously shown.”

 We have rewritten a line in the last paragraph of the Introduction section trying to make clear that we use the same database. It now reads (page 4, lines 79-82): “This work uses a sample of students that was used in an earlier work aiming to study the factors associated with an unhealthy diet [14]. That study analysed diet quality through the use of an index, while the current work has adopted a different approach, using new variables.” Please, if you think we can explain this better, let us know.

On the other hand, and regarding Table 1, we have added a line in the Results section. This now reads (page 9, lines 183-189): “Table 1 shows the characteristics of the population by student gender. The students’ age, whether they cooked (or not) for themselves, and the degree course were previously shown in our earlier work [14]. In the present study, we also included the location of the family home, but using three categories that consider the size of the family home town. The socioeconomic status of the family is a new variable.”. Please, if you think we can write or express this issue in a better way, let us know.

---

## [Editor Report · Decision Letter 2]

26 Dec 2019

Social determinants of food group consumption based on Mediterranean diet pyramid: a cross-sectional study of university students

PONE-D-19-14439R2 ( Dear Dr. Martinez-Lacoba, We are pleased to inform you that your manuscript has been judged scientifically suitable for publication and will be formally accepted for publication once it complies with all outstanding technical requirements.

If your institution or institutions have a press office, please notify them about your upcoming paper to enable them to help maximize its impact. If they will be preparing press materials for this manuscript, you must inform our press team as soon as possible and no later than 48 hours after receiving the formal acceptance. Your manuscript will remain under strict press embargo until 2 pm Eastern Time on the date of publication. For more information, please contact onepress@plos.org.Berta SchnettlerAademic EditorPLOS ONEAdditional Editor Comments (optional):

---

## [Editor Report · Acceptance letter]

16 Jan 2020

PONE-D-19-14439R2 

Social determinants of food group consumption based on Mediterranean diet pyramid: a cross-sectional study of university students 

Dear Dr. Martinez-Lacoba:

I am pleased to inform you that your manuscript has been deemed suitable for publication in PLOS ONE. Congratulations! Your manuscript is now with our production department. 

With kind regards,

on behalf of

Dr. Berta Schnettler 

Academic Editor

PLOS ONE